# Novel Type of Tetranitrosyl Iron Salt: Synthesis, Structure and Antibacterial Activity of Complex [FeL’_2_(NO)_2_][FeL’L”(NO)_2_] with L’-thiobenzamide and L”-thiosulfate

**DOI:** 10.3390/molecules27206886

**Published:** 2022-10-14

**Authors:** Nataliya A. Sanina, Arina A. Starostina, Andrey N. Utenyshev, Pavel V. Dorovatovskii, Nina S. Emel’yanova, Vladimir B. Krapivin, Victor B. Luzhkov, Viktoriya A. Mumyatova, Anastasiya A. Balakina, Alexei A. Terentiev, Sergey M. Aldoshin

**Affiliations:** 1Federal Research Center of Problems of Chemical Physics and Medicinal Chemistry, RAS, 1, Acad. Semenov Av., 142432 Chernogolovka, Russia; 2Faculty of Fundamental Physical-Chemical Engineering, M.V. Lomonosov Moscow State University, 1/51 Leninskie Gory, 119991 Moscow, Russia; 3Scientific and Educational Center in Chernogolovka, Medical-Biological Institute, Moscow Regional State University, 24, Vera Voloshina Street, 141014 Mytischi, Russia; 4National Research Centre “Kurchatov Institute”, 1 Acad. Kurchatov Square, 123182 Moscow, Russia

**Keywords:** nitric oxide donors, tetranitrosyl iron complex, thiobenzamide, X-ray diffraction, IR spectroscopy, TDDFT, molecular modeling, antibacterial activity, biofilm, cytotoxicity

## Abstract

In this work a new donor of nitric oxide (NO) with antibacterial properties, namely nitrosyl iron complex of [Fe(C_6_H_5_C-SNH_2_)_2_(NO)_2_][Fe(C_6_H_5_C-SNH_2_)(S_2_O_3_)(NO)_2_] composition (complex **I**), has been synthesized and studied. Complex **I** was produced by the reduction of the aqueous solution of [Fe_2_(S_2_O_3_)_2_(NO)_2_]^2−^ dianion by the thiosulfate, with the further treatment of the mixture by the acidified alcohol solution of thiobenzamide. Based on the structural study of **I** (X-ray analysis, quantum chemical calculations by NBO and QTAIM methods in the frame of DFT), the data were obtained on the presence of the NO…NO interactions, which stabilize the DNIC dimer in the solid phase. The conformation properties, electronic structure and free energies of complex **I** hydration were studied using B3LYP functional and the set of 6–31 + G(d,p) basis functions. The effect of an aquatic surrounding was taken into account in the frame of a polarized continuous model (PCM). The NO-donating activity of complex **I** was studied by the amperometry method using an “amiNO-700” sensor electrode of the “inNO Nitric Oxide Measuring System”. The antibacterial activity of **I** was studied on gram-negative (*Escherichia coli*) and gram-positive (*Micrococcus luteus*) bacteria. Cytotoxicity was studied using Vero cells. Complex **I** was found to exhibit antibacterial activity comparable to that of antibiotics, and moderate toxicity to Vero cells.

## 1. Introduction

Over the past few decades, the world has become increasingly concerned about the danger of infections caused by pathogenic microorganisms. Current antibiotics are effective in the treatment of many infectious diseases, but their widespread use has led to the emergence of various multidrug-resistant strains and superbugs. There is an urgent need to develop new drugs and materials that can fight drug-resistant bacteria. New strategies and approaches are necessary to expand the spectrum of functional activity of new drugs without the development of drug resistance. One such strategy is chemotherapy with nitric oxide (“NO therapy”) based on a huge experimental material demonstrating NO’s effect both on the development of pathological processes [1] and on the possibility of their correction by chemotherapeutic methods [2].

The implementation of such a strategy includes the chemical synthesis of small molecules, and the use of experimental physical–chemical methods and methods of quantum chemical calculations to study their structure and properties in the solid phase and in solutions.

Being effective low-toxic exogenous nitric oxide (NO) donors complementary to a human organism, low-molecular dinitrosyl complexes of tetrahedral iron involving two nitrosyl and two sulfur-containing ligands, as structural analogs of the active sites of dinitrosyl [1Fe-2S] proteins, are promising for practical application in medicine, which is the reason for the increasing interest in fundamental studies of their structural features [3,4,5]. As follows from our investigations, the activity of dinitrosyl complexes of tetrahedral iron (DNICs) and their dimers (Figure 1A,B) as antibacterial agents is comparable and/or higher than that of antibiotics used in clinics [6,7,8,9,10].

As a basic mechanism of the antimicrobial activity of NO donors of this class, the induction of oxidative–nitrosative stress is considered, which involves the interaction of NO generated by the nitrosyl complexes with oxygen and its active forms, thus leading to chemically active nitrogen forms arising that damage macromolecules, primarily, DNA [11]. Concurrently, nitric oxide can cause the modification of thiol-containing amino acids in proteins, thus changing their activity [12]. We have demonstrated that compounds of this family of NO donors cause nitrosylation and impair the functions of the NF-kB transcription factor [13]. 

In the majority of works, aliphatic and aromatic thiols are used as ligands for the synthesis of DNICs and their dimers. Thioamide-based DNICs are virtually unknown. Due to a unique electron donor nature of heterocyclic thioamides, they possess a wide range of properties that are of great practical application in various fields of clinical chemistry and pharmacology. Given that thioamides have a higher biologic activity, including a higher antibacterial activity, than the corresponding amides [14,15,16] and some of them are actively used as antituberculosis drugs [17] and antimicrobial agents [18,19,20], the high activity of DNIC with thioamides can be expected, which will be comparable to the activity of clinical antibiotics in the absence of pathogen resistance. In this regard, thioamides, on the one hand, are of interest as sulfur-containing ligands for the synthesis of nitrosyl iron complexes, i.e., two potentially functional blocks, NO groups and thioamides, can be combined in one molecule to yield a new family of prospective antibacterial compounds. On the other hand, it is interesting to investigate the bond’s nature and the nature of intermolecular interactions, as well as the electronic structure of the {Fe(NO)_2_}^9^ dinitrosyl fragment in the thioamide DNICs, which continues to be the subject of active discussion since general acceptance of the formalism [21] to the present time [22].

In this work, a new nitrosyl iron complex of [Fe(C_6_H_5_C-SNH_2_)_2_(NO)_2_][Fe(C_6_H_5_C-SNH_2_)(S_2_O_3_)(NO)_2_] composition (complex **I**) (Figure 1C) was first synthesized. Experimental and theoretical investigations of its structure and antibacterial activity were performed, as well as correlations of “structure-activity” analyzed, taking into account the nature of intermolecular interactions that cause the emergence of the NO…NO contacts.

## 2. Results

### 2.1. Synthesis of Complex ***I***

Complex **I** was synthesized according to Figure 1. The dianion of tetranitrosyl iron dithiosulfate, [Fe_2_(S_2_O_3_)_2_(NO)_4_]^2−^, forms a water-soluble anion of dinitrosyl iron dithiosulfate in the presence of S_2_O_3_^2−^ in an aqueous solution. The structure of this intermediate was determined earlier by the EPR spectroscopy method [23,24]. 

In this reaction, incomplete replacement of the thiosulfate ligands by the thiobenzamide ones occurs in the reduced mononuclear dinitrosyl iron trianion in an aqueous acidic solution.

### 2.2. Single Crystal X-ray Studies of Complex ***I***


A general view of complex **I** is shown in Figure 2. The bond lengths and valence angles in the coordination centers are shown in Table 1. In the **I** cation, an intramolecular hydrogen bond of the N-H…N type forms, with parameters shown in Table 2.

### 2.3. Quantum Chemical Calculations of Intermolecular Interactions in Complex ***I***

To reveal the nature of pair intermolecular interactions and taking into account that the cation and anion in complex **I** form a dimeric associate, we performed calculations of tetrameric associates, which correspond to two different pair intermolecular interactions along two different axes in the crystal, using QTAIM based on topological analysis of the electron density distribution function ρ(r) [25]. Figure 3 presents molecular graphs of tetrameric associates obtained from this method. Table 3 shows critical points and the calculated energy of the bonds in the tetrameric associate *I*.

NBO analysis data of the orbital interactions of the NO ligands participating in the O…O contacts of the tetrameric associate *I* and of the orbital interactions in the tetrameric associate *II* inducing the NO…NO contact are presented in Table 4 and Table 5, respectively.

### 2.4. Molecular Modeling of Dissociation and Decomposition Products of Complex ***I***

The molecular structure and electronic characteristics of dissociation and decomposition products of NO donors of this family in water essentially affect their binding with biomolecular targets. To study the structure of **I** in an aqueous medium, quantum mechanical calculations are necessary. 

In an aqueous solution, **I** decomposes to cationic and anionic complexes [Fe(NO)_2_(TBA)_2_]^+^ (***1***) and [Fe(NO)_2_(TBA)(S_2_O_3_)]^−^ (***2***), where TBA is thiobenzamide C_6_H_5_C(NH_2_)=S. The chemical structure of these molecules is presented in Figure 4. Conformation mobility of compounds ***1***, ***2*** is due to the rotation around the Fe-S, S=C and C-C bonds (Figure 2). Rotation around the Fe-S bonds (α angles) and S=C bonds (β angles) can be presented as a set of g+, g− and tr conformers. Rotation around the C-C bond (γ angles) yields g+ and g− conformers. For complex ***1***, all symmetric conformations were considered, as well as asymmetric ones with the most energetically favorable orientation of the ligands (Appendix A). For complex ***2***, all conformations related to TBA ligand orientation were considered (Appendix A).

The rotation area of side chains at angles α, β and γ is given in Appendix A by a trio of conformers in the brackets at each discovered structure. At the same time, the specified three conformers give only a qualitative description of the structure type, while the optimized angles deviate from the ideal values +60, −60 and 180° for the corresponding conformations. The total internal energy of conformers E_0,w_ was calculated as E_0,w_ = E_el,w_ + ZPVE (E_el,w_ is total electron energy including free solvation energy, ZPVE is zero vibration energy).

### 2.5. NO Donor Activity of Complex ***I***

It was found that in 0.1% aerobic aqueous DMSO solutions, complex **I** generates NO without additional activation (photo-, thermo- or redox agents). The NO release during the decomposition of the complex is observed ~10 s after dissolution. Figure 5 presents the time dependencies of NO generated by complex **I**.

### 2.6. Antibacterial Activity of Complex ***I***

Table 6 shows the minimum inhibitory concentration (MIC) values for **I** and its ligands, thiobenzamide and sodium thiosulfate, as well as some known antibiotics. Complex **I** was found to inhibit the growth of both gram-positive (*M. luteus*) and gram-negative (*E. coli*); its growth inhibiting activity is close to that of kanamycin and streptomycin. Both ligands did not exhibit growth inhibiting effects at concentrations up to 1 mM.

Complex **I** has also been investigated for the ability to inhibit the biofilm formation of *M. luteus*. As is seen from Figure 6, **I** at a concentration corresponding to MIC inhibits the biofilm formation by appr. 28%, which is comparable to the effects of ampicillin and ceftriaxone, and more efficient than that of kanamycin and streptomycin.

### 2.7. Cytotoxicity Activity of Complex ***I***

Figure 7 presents the results of studying the cytotoxic action of complex **I** and its ligands; IC_50_ doses calculated on the basis of the curves are shown in Table 7. None of the ligands exerted apparent effects on cell viability at a concentration of 500 μM or lower.

## 3. Discussion

Thioamides, being the thioanalogues of amides of carboxylic acids, are usually described by structure ***a*** (Figure 8). The structure of these compounds is favorable for donating the lone electron pair of the nitrogen atom to the π-antibonding orbital of the double bond, thus stabilizing the molecule planar structure. Due to such a structure, *N*-monosubstituted and asymmetrical *N*-disubstituted thioamides should possess a capability of geometrical isomerism, i.e., they can exist in one of the two (or in both) forms ***a*** and ***b***. Theoretically, for primary and secondary thioamides, there is a possibility to transform into thioamide form ***d***. The existence of ***d*** ↔ ***c*** equilibrium has often been suggested, based mainly on chemical data; however, X-ray analysis, and IR and NMR spectroscopy demonstrated no measurable concentrations of ***d*** both in solutions and the solid state. At very high values of α-CH-acidity, the existence of ***f*** tautomer was observed.

The vibration of the C=S group in IR spectra has a frequency ν(CS) equal to 1275 cm^−1^ [26]. In the IR spectra of thioamides, the characteristic band of an individual valence vibration of the C=S group is normally absent, in contrast to thioureas, for example (1100–1400 cm^−1^) [27]. Vibrations of the S-H group are known to lie in the range of 2500–2600 cm^−1^, and they are not observed in the IR spectrum of the initial thiobenzamide either (Appendix A).

In the IR spectrum of complex **I**, sin-phase vibrations of the NO group (νNO-sym) are observed at ~1793 cm^−1^, and antiphase vibrations (νNO-asym) at ~1723 cm^−1^. The intensity of the bands of the NO groups’ valence vibrations is higher than that for other bands in the spectrum of **I**. The difference between νNO-sym and νNO-asym (ΔνNO) is 70 cm^−1^. This value of splitting of the νNO vibration doublet suggests non-equivalence of the NO groups, as with dimers of 1,2,4-triazolethiolyls of µ-NCS structural type [10,28], which appears in the inconsiderable change in the N-O and Fe-N bond lengths within the margin of error. 

The molecular structure of **I** is shown in Figure 2. Coordination centers Fe(1) and Fe(2) have a distorted tetrahedral coordination of the iron atoms. The Fe-N bond lengths in the cation and anion are practically identical and equal to: Fe(1)-N(1) = 1.689(3) Å, Fe(1)-N(2) = 1.681(3) Å and Fe(2)-N(5) = 1.688(3) Å, Fe(2)-N(6) = 1.681(3) Å, respectively. The bond length Fe(1)-S(1) 2.3394(9) Å in the Fe(1) coordination center is elongated by 0.0494 Å as compared to the bond length Fe(1)-S(2) 2.2900(9) Å, and by 0.0152 Å as compared to the bond length Fe(2)-S(3) = 2.3242(9) Å in the Fe(2) coordination center. The Fe(2)-S(4) bond length is the least of all the Fe-S bonds (2.2650(9) Å). Formerly, in the structure of the neutral mononuclear iron complex with 1,2,4-triazole-3-thione, a considerable non-equivalence of the Fe-S bonds was revealed, 2.3015(2) and 2.3219(2) Å (the difference 0.02 Å), which was due to different types of chemical binding between the iron and sulfur atoms, both at room and low (100 K) temperatures [29,30]. In the studied compound, this tendency is enhanced (the difference in the lengths of the Fe-S bonds is 0.0494 Å in the Fe(1) coordination center and 0.0592 Å in the Fe(2)coordination center). The bond lengths of the NO groups practically do not differ from the previous studies [31,32,33,34], but they differ slightly from each other: in the Fe(1) center: O(1)-N(1) = 1.170(4) Å, O(2)-N(2) = 1.170(3)Å, and O(3)-N(5) = 1.166(3) Å, O(4)-N(6) = 1.174(3) Å in the Fe(2) center. The carbon–sulfur bond lengths of the thioamide ligand are S(1)-C(1) = 1.697(3) Å, S(2)-C(8) = 1.717(3) Å and S(3)-C(15) = 1.702(3) Å, which correspond to similar bonds in mononuclear complexes [30], and slightly exceed the length of S=C equal to 1.684 Å [35], but they are shorter than the single bond length S-C 1.719 Å in ligands with a thiol form. Thus, it can be concluded that the ligand coordinates the iron atoms in the thion form (structure ***c*** in Figure 8).

The lengths of the amide C-N bonds in the three ligands are reduced (1.304 Å average) due to the amide n–π*conjugation, but remain longer than double C=N bonds.

In the complex anion, unlike a cation, an intramolecular hydrogen bond of the N-H…O type is realized, and between the cation and the anion, an intermolecular hydrogen bond of the N-H…O type is also realized (Table 2).

The oxygen atom O(1) of the NO group of the complex cation has a contact with the oxygen atom O(3A) {x + 1, y, z} of the NO group of the complex anion; the interatomic distance O(1)… O(3A) is equal to 2.787 Å (Figure 9).

The crystal structure of complex **I** is represented by alternating layers of cationic and anionic complexes. A fragment of the crystal structure of **I** is shown in Figure 10.

The shortest distances between the iron atoms in the crystalline packing of **I** are Fe(1)… Fe(1)* {−x + 2, −y + 1, −z + 1} and Fe(1)… Fe(2)* {−x + 1, −y + 1, −z + 1} with lengths of 5.371(3) and 5.815(3) Å, respectively. In the crystal, the network of intermolecular hydrogen bonds of N-H…O type forms (Figure 11). All three oxygen atoms of the S_2_O_3_^2−^ group are involved in the formation of these intermolecular hydrogen bonds (Table 1). It should be noted that oxygen atoms O(5) and O(6) are involved in the formation of two hydrogen bonds of N-H…O type, one intramolecular and one intermolecular.

In this work we were mostly interested in the NO…NO interaction (the presence of critical points of (3, −1) type), which may stabilize the structure of the DNIC dimer. In addition, as stated earlier [36], involvement of the NO groups’ orbitals in the intermolecular NO…NO interactions led to quenching of the orbital moment, while unquenched orbital moment of NO led to abnormal magnetic properties.

Intermolecular interactions along the first direction (the associate *I*) are rather complicated, as follows from Figure 3 where molecular graphs of the tetrameric associates are presented. QTAIM yields nine critical points along the bond lines of the atoms of two different dimeric molecules, and they are located almost symmetrically with regard to CP1 that is related to the NO…NO interaction we are interested in. Evidently, these bonds are related to rather weak intermolecular interactions; therefore their energy E_A-B_ can be estimated from the available correlation formula [37] via potential energy density in the corresponding critical point ve(r): E_A-B_ ≈ 0.5ve(r). 

The main contribution (11.0 kcal/mol) is made by hydrogen bonds between the oxygen atoms of the S_2_O_3_ groups and the hydrogen atoms of the NH_2_ groups of the ligands; the NO…NO contacts are much weaker but are slightly higher than the others. We can see that the NO groups form intermolecular contacts with the sulfur atoms of the S_2_O_3_ group at the expense of the oxygen atom. Generally, the S_2_O_3_ ligand easily forms contacts both at the expense of the oxygen, and the sulfur atoms. 

We have performed NBO analysis of the tetrameric associate *I* to reveal the nature of the O…O intermolecular interaction (CPP1). The bond order and Wiberg index for this bond are 0.0004 and 0.0001, respectively. To understand whether there is quenching of the orbital moment due to the involvement of NO π-orbitals in intermolecular NO…NO contacts [36], the second order interactions were analyzed. Unexpectedly, no secondary interactions between the considered NO ligands were determined. The presence of the electron density in the CP1 critical point can be explained by delocalization of the electron density due to multiple secondary interactions of the NO groups with the sulfur and iron atoms of the neighboring molecule (Table 4).

The structure of tetrameric associate *II* along the other axis in the crystal differs essentially from that of *I*. First, from QTAIM analysis there is only one critical point (3, −1) corresponding to the intermolecular contact, and it lies on the bond length of two oxygen atoms in NO…NO. This bond energy calculated from the Espinosa–Lecomte formula is 2.48 kcal/mol. From NBO analysis of this tetrameric associate, the Wiberg index and the bond order are 0.0019 and 0.0023, respectively, which are higher than for associate *I*. We also analyzed secondary interactions to reveal the nature of the donor and acceptor orbitals of this bond (Table 5).

It can be seen that NBO analysis confirms the presence of NO…NO interaction in this case. Each oxygen atom donates the electronic density of the lone electron pair on π* (NO) of the neighboring molecule. This donation occurs easier along O → O-N with an 1800 angle than with a 900 angle. As we mentioned, two NO ligands are located perpendicular to each other, such interaction being called T-like according to the classification in [38]. Moreover, electron exchange occurs between the π*-orbitals of the NO ligand. Thus, in complex **I** there is an NO…NO contact provided by NO π-orbitals, which leads to quenching of the orbital moment.

According to calculations (see the results of molecular modeling of complex **I**), four symmetrical conformations of cation ***1*** (c2, c5, c15, c16) appeared to be the most stable (within 3 kJ/mol), as well as four asymmetrical ones, in which the orientation of the ligands corresponds to the combination of conformers c2, c5 and c12. For all low-lying conformers, the characteristic values of α and β angles correspond to the values (g+, tr, …). For anion ***2***, two conformations (c1 and c2) appeared to be the most stable, which are characterized by the deviation of the benzene ring from the S–C–N plane. The conformation observed in the crystal packing of the ion pair ***1***… ***2*** of complex **I**, has a relative energy of 5.9 kJ/mol for ***1*** (Appendix A) and 1.24 kJ/mol for ***2*** (Appendix A). Figure 12 shows overlapping of the calculated conformations ***1*** and ***2***. Optimized atomic coordinates for the conformers from Appendix A are presented in the Appendix A. Overall, the calculations showed a wide variety of thermally available structures for molecules ***1*** and ***2***.

The results of the conformational analysis are taken into account in the subsequent modeling of binding ***1*** and ***2*** with active centers of targets in vivo. It is also concluded that when choosing a method for binding calculations, one should focus on methods of flexible ligand docking.

Figure 5 presents the time dependencies of NO generated by complex **I**. The kinetic dependence for complex **I** goes to a “plateau” in 300 s, and the amount of NO formed (~5.0 nM) does not decrease by 500 s of the experiment, which indicates the formation of long-lived nitrosyl intermediates of this complex, which may be responsible for prolonged NO donor activity. Based on the data of X-ray analysis and molecular modeling for complex **I**, it can be assumed that mononuclear dinitrosyl [1Fe-2S] intermediates are responsible for prolonged NO release, as with antibacterial DNIC dimers with 5-nitrophenyl-4 H-1,2,4-triazole-3-thiolyls [10], which were studied earlier.

The antibacterial effect of **I** (see Figure 6) is similar to or higher than that of the comparator drugs, kanamycin and streptomycin, against gram-positive bacteria *M. luteus*. For gram-negative bacteria *E. coli*, the MIC value for **I** was two times higher. The difference in the sensitivity of gram-negative and gram-positive bacteria can be explained by differences in the cell wall architecture. Gram-positive microorganisms are known to have one cell membrane consisting of peptidoglycan. The cell wall of gram-negative bacteria has a more complex organization, having internal, plasmatic and external membranes enriched with lipopolysaccharides. In addition, the cell wall of gram-negative bacteria has many dense intermediate layers. Such an arrangement of the cell wall forms a selective barrier that protects the cell from external influences, including antibiotic therapy [39,40,41].

Complex **I** exhibited a cytotoxic effect on Vero cells with IC_50_ of appr. 90 μM (Table 7). It should be noted that at a concentration of 62.5 μM, i.e., a concentration at which complex **I** inhibits the growth and biofilm formation of *M. luteus*, a decrease in cell viability was apparently low (appr. 25%).

Ligands present in the structure of complex **I** do not exhibit antibacterial activity against both bacterial strains studied (Table 6). No remarkable cytotoxicity to Vero cells was also detected for the ligands at concentrations up to 500 μM (Figure 7). Thus, the antibacterial and cytotoxic effects of the complex are determined by the overall structure of the compound, but not by the ligands comprising its structure.

Comparison of the antibacterial, cytotoxic and NO-donating properties of complex **I** and DNICs with thiolyls studied earlier [9,10,42,43,44] leads to some important points.

In our previous work, sixteen nitrosyl iron complexes were found to inhibit the growth of gram-positive bacteria and, with less efficacy, gram-negative bacteria. Most of these complexes were efficient inhibitors of biofilm formation [9]. Three DNIC dimers with 5-(nitrophenyl)-4-H-1,2,4-triazole-3-thiolyls exhibited an equal growth-inhibiting effect on *M. luteus* and *E. coli* and inhibited biofilm formation at a wide range of concentrations [10]. As was found for complex **I**, DNICs with nitrothiophenols or dimethylthiourea [9] and 5-(nitrophenyl)-4-H-1,2,4-triazole-3-thiolyls [10], the antibiofilm activity of DNICs is comparable to that of antibiotics. Thus, DNICs of different structures exert antibacterial activity.

The cytotoxicity of DNICs varies in a wide range, but comparison with corresponding MICs demonstrates that many DNICs exhibit higher activity on bacteria than on eucaryotic cells. For complex **I** (Table 6) and DNICs with 3-nitrothiophenol, dimethylthiourea [9] and 5-(nitrophenyl)-4-H-1,2,4-triazole-3-thiolyls [10], IC_50_ values for Vero cells exceeds corresponding MICs for *M. luteus*. IC_50_ for DNICs with 5-(nitrophenyl)-4-H-1,2,4-triazole-3-thiolyls are also higher than MICs for *E. coli* [10]. Conversely, DNIC with 4-nitrothiophenol exerted cytotoxicity with IC_50_ lower than MICs for both *M. luteus* and *E. coli* [9]. Based on these in vitro data, one may conclude that different DNICs can be considered as antibacterial agents except 4-nitrothiophenol-bearing DNIC.

The NO generation kinetics of all DNICs under discussion have a plateau, which demonstrates the prolonged NO release. The maximum level of NO generated by complex **I** is valued at 5 nM (Figure 5). Other DNICs generate NO with greater levels at their plateau. The maximum generated NO reached a value of ~60 nM for the DNIC dimer with 3-nitrothiophenol [42], ~25 nM for the DNIC dimer with 4-nitrothiophenol [8] and 10 to 20 nM for DNICs with 5-(nitrophenyl)-4-H-1,2,4-triazole-3-thiolyls [10].

From these data it can be seen that both the antibacterial activity and cytotoxicity of DNICs cannot be attributed solely to their NO generation features. It is important to note that biological activity can also not be attributed to ligands of DNICs since the antibacterial and cytotoxic effects of ligands were not detectable (Table 6 and Table 7).

Thus, the antibacterial properties of studied DNICs are not determined by their ligands or by the amount of generated NO, but rather by the overall structure and physicochemical properties. Since corresponding ligands do not exert the biological activity of DNICs, it can be assumed that intermediates formed upon DNIC decomposition can contribute to the biological effects of NO-donating DNICs. Further design of antibacterial agents of this family should be directed towards the synthesis of complexes with prolonged NO generation, and a detailed analysis of the structure of the resulting long-lived nitrosyl intermediates can give insights into the mechanisms of their antibacterial action.

## 4. Materials and Methods

All operations for the synthesis of complex **I** were carried out in an inert atmosphere. Beforehand, argon (technical) was purged for 30 min through the bidistilled water used for the synthesis. For the synthesis of Na_2_[Fe_2_(S_2_O_3_)_2_(NO)_4_]·4H_2_O, according to the procedure in [43], the following reactants were used: FeSO_4_·7H_2_O, Na_2_S_2_O_3_·5H_2_O (Sigma-Aldrich, St. Louis, MO, USA). NO was prepared by the reaction of iron (II) sulfate and sodium nitrite solutions in an acidic medium. For this, FeSO_4_·7H_2_O, NaNO_2_ (pure grade) and HCl (pure grade, conc.) were used. 

Analysis of C, H, N, S, O elements in complex **I** was performed on a CHNS/O elemental analyzer “Vario El cube”. Fe was detected on the atomic absorption spectrometer “AAS-3”. A sample of polycrystals of complex **I** was preliminarily mineralized. IR spectra were recorded on a Fourier spectrometer (Bruker ALPHA, Ettlingen, Germany) in the frequency range of 400–4000 cm^−1^ in the ATP mode at room temperature.

### 4.1. Synthesis

Thiobenzamide powder (97%, Sigma-Aldrich, St. Louis, MO, USA) (0.48 g, 3.5 mmol) was dissolved in 0.5 mL of concentrated HCl and 5 mL of methanol. The acidified solution was stirred without heating and filtered. A dry mixture of 0.38 g (0.4 mmol) of Na_2_S_2_O_3_·5H_2_O and 0.44 g (0.77 mmol) of Na_2_[Fe_2_(S_2_O_3_)_2_(NO)_4_]·4H_2_O was dissolved in 5 mL of water and filtered. The filtrate was purged for 1–2 min in an inert gas current. To the filtrate of the acidified thiobenzamide solution in the argon current, the filtrate of the mixture of sodium thiosulfate and nitrosyl iron complex was quickly poured, with the inert gas still passing through the solution. The instantaneous precipitate was filtered, and the filtrate was stored in a closed flask at +5 °C. After 3 weeks, the formed single crystals of complex **I** as red-brown thin needles of ~0.2 mm were filtered and dried in argon.

For C_21_H_21_N_7_O_7_S_5_Fe_2_: Found, % C 33.54, H 2.80, N 12.95, O 14.79, S 21.17, Fe 14.75. Calculated, % C 33.40, H 2.78, N 12.98, O 14.83, S 21.22, Fe 14.79.

IR spectrum of **I** (cm^−1^): 3379 (v.w.), 3030 (w), 1793 (v.s.), 1723 (v.s), 1467 (m.), 1259 (m.), 1113 (m.), 1117 (m.), 997 (s.), 856 (m.), 760 (m.), 701 (m.), 680 (m.), 631 (m.), 524 (m.), 421 (m.), νNO = 1793 cm^−1^, 1723 cm^−1^.

### 4.2. X-ray Analysis

X-ray study of single crystals of **I** was performed at 100 K on a “Belok” beamline diffractometer of the National Research Center “Kurchatov Institute” (Moscow, Russia) using a Rayonix SX165 detector at λ = 0.96990 Å. The data were indexed and integrated using an iMOSFLM utility in CCP4 program [44], and then scaled with absorption correction using the Scala program [45]. Crystallographic data and main refinement parameters are shown in Table 8. The structure was solved by the direct method [46]. Positions and temperature parameters of non-hydrogen atoms were refined in the anisotropic approximation by full-matrix least-squares method [46]. Positions of all hydrogen atoms were calculated geometrically and then refined by imposing the restrictions of the “riding” model [46]. All calculations were performed using SHELXTL program complex [46]. The crystalline structure was deposited with the Cambridge Structural Database, CCDC, Deposition Number 2182839, and it can be freely received at www.ccdc.cam.ac.uk/data_request/cif (accessed on 12 July 2022).

### 4.3. Quantum Chemical Calculations

We performed quantum chemical calculations of tetrameric associates *I* and *II* using GAUSSIAN-09 program [47] without optimization of their geometrical structure, in tpssh/6–31G* approximation. The AIMAll program package (Version 15.05.18) [48] was used to analyze the wave function by QTAIM methods. The NBO computational procedure implemented in the Gaussian 09 program was used.

Conformation properties, electronic structure and free energies of complex I hydration were studied by means of quantum mechanical calculations by density functional theory method using B3LYP functional and a set of basis functions 6–31 + G(d,p). The effect of water surrounding was taken into account in the frame of polarized continuum medium model (PCM) [49] and SMD [50]. A license program Gaussian-09 was used [47].

### 4.4. Amperometric Determination of NO Generation

To determine NO amount generated by complex **I** in solution, a sensor electrode “amiNO-700” of “Nitric oxide measurement system inNO-T” (Innovative Instruments, Inc., Tampa, FL, USA) was used. All the experiments were performed in aerobic 0.1% aqueous DMSO solution at 25 °C and pH 7.0. NO concentration was recorded for ~500 s (with 0.2 s pace). DMSO was purified according to the technique in [51]. Commercial buffer Hydrion (Sigma-Aldrich, N 239089, St. Louis, MO, USA) was used (pH of the solutions was measured using a membrane pH-meter “HI 8314” (HANNA Instruments, Vöhringen, Germany)). A standard 100 μM NaNO_2_ aqueous solution was added to the solution. The mixture of 0.12 M KI and 2 mL of 1 M H_2_SO_4_ in 18 mL of water was used for calibration of the electrochemical sensor.

### 4.5. In Vitro Antibacterial Activity

Antibacterial activity of **I** was studied by the serial dilution method [10]. Laboratory cultures of gram-negative *Escherichia coli* bacteria (strain BB) and gram-positive Micrococcus luteus bacteria (strain 21/26) were used as bacterial agents. The bacteria were dispersed into 96-well flat-bottomed plates in the LB incubation medium (Peptone 1%, yeast extract 0.5%, NaCl 1%, glucose 0.1%, pH 6.8–7.0) of 100 μL each, and 100 μL of consecutive double dilutions of the test substances were introduced. The initial solution of the compounds was prepared in DMSO at a maximum concentration of 200 mM. Final concentrations of compounds in the samples ranged from 1000 μM to 0.47 μM. The final maximum concentration of DMSO in the samples was 10%. This concentration of DMSO does not affect the viability of the bacterial agent. The final concentration of the bacterial agent in the well was 5 × 10^5^ CFU/mL. 

As comparison drugs, antibiotics ampicillin (JSC “Sintez”, Kurgan, Russia), kanamycin (JSC “Sintez”, Kurgan, Russia), streptomycin (JSC “Biochemist”, Saransk, Russia) and ceftriaxone (JSC “Biochemist”, Saransk, Russia), dissolved in the LB incubation medium, were used. Final concentrations of antibiotics ranged from 1000 μM to 0.49 μM. Samples were incubated at 37 °C for 24 h after administration of the complex and antibiotics. Control samples were grown in the LB medium or in the LB medium with 10% DMSO at 37 °C. Negative control was grown in the LB medium at 4 °C. The minimum inhibitory concentration of the compound that inhibits the growth of microorganisms (MIC) was defined as the minimum concentration at which there was no visible growth of microorganisms compared to the control samples.

To obtain biofilms, gram-positive Micrococcus luteus bacteria (strain 21/26) were cultured in a 96-well plate at a concentration of 5 × 10^6^ CFU/mL at 37 °C. For determination of the background staining, samples with LB medium without bacteria were used. Control samples contained bacteria in either LB medium or LB medium with solvent (10% DMSO). All compounds and comparator drugs were used at concentrations corresponding to MICs. After 24 h of incubation in the presence of studied compounds, the medium with planktonic cells was removed, the wells were washed with sterile PBS buffer (137 mM NaCl, 2.68 mM KCl, 4.29 mM Na_2_HPO_4_, 1.47 mM KH_2_PO_4_, pH 7.4) and dried at 37 °C for 60 min, then stained with 0.1% solution of crystal violet for 15 min. After staining, the wells were washed with water and dried. The dye bound to biofilms was dissolved in 95% ethanol, and optical density of the resulting solutions was determined photometrically at a wavelength of 570 nm. The optical density of the solution in control samples with the bacteria suspension without the test complexes was taken as 100%. Background staining of the substrate (empty wells) was subtracted from all samples.

### 4.6. In Vitro Cytotoxicity Assay

#### 4.6.1. Cell Culture

As a model, a cell culture of Vero (African green monkey kidney epithelial cells) was used. Cells were grown DMEM incubation mediums supplemented with 10% fetal calf serum, penicillin (100 U/mL) and streptomycin (100 µg/mL). The cells were cultured at 37 °C in a humidified atmosphere with 5% CO_2_.

#### 4.6.2. MTT Assay

To assess the cytotoxic effect of **I**, the MTT test method was used [52]. Vero cells (5 × 10^3^ cells) were plated into 96-well plates in 100 µL aliquots and allowed to adhere for 24 h. All studied compounds were dissolved in DMSO and used immediately. The initial concentration of all compounds was 500 mM, and serial dilutions were prepared just before use. Incubation medium in all samples was aspirated and replaced by medium with studied compounds. DMSO concentration in all samples was 0.1%, the final concentration of complex **I** and its ligands ranged from 7.81 to 500 µM. After 72 h of incubation, 10 µL of MTT solution (5 mg/mL) was added and plates were incubated for an additional 4 h at 37 °C, then the media removed and MTT formazan crystals were dissolved in 100 µL of 100% DMSO. The absorption of the samples was measured at 570 nm (reference wavelength 650 nm) on Spark 10 M multimode microplate reader (Tecan, Männedorf, Switzerland).

## 5. Conclusions

According to X-ray analysis data, a new nitrosyl iron complex, namely a salt [FeL’_2_(NO)_2_][FeL’L”(NO)_2_] with L’-thiobenzamide and L”-thiosulfate (complex **I**), was obtained by incomplete replacement of thiosulfate ligands by thiobenzamide in [Fe(S_2_O_3_)_2_(NO)_2_]^3−^ anion in an acidified water–alcohol solution. The Fe atoms are coordinated by the L′ ligand in thionic form, the cation and anion are hydrogen bonded into a dimeric associate. Calculations of tetrameric associates (by QTAIM and NBO methods) corresponding to two different pair intermolecular interactions along two different axes in the crystal **I** showed the presence of a critical point corresponding to the intermolecular contact on the bond line of two oxygen atoms in NO… NO (the energy of this bond is 2.48 kcal/mol). Each oxygen atom easily donates the electron density of the lone electron pair to the π* (NO) of the neighboring molecule in the direction of O → O-N with an angle of 1800 of the tetrameric associate of **I**, causing its stability in the solid phase. 

Quantum mechanical calculations of the dissociation products of complex **I** in aqueous solution, cationic [FeL’_2_(NO)_2_]^+^ and anionic [FeL’L”(NO)_2_]^−^ components, have been performed for subsequent modeling of their binding to active centers of biological targets. It has been shown that there is a variety of thermally accessible structures for cation and anion, which is consistent with the experimental amperometry data on the NO-donating activity of complex **I**.

The resulting complex **I** was tested against gram-negative bacteria *Escherichia coli* (strain BB) and gram-positive bacteria *Micrococcus luteus* (strain 21/26). The DNIC dimer showed an inhibitory potential with a value of 62.5 mM against gram-positive *M. luteus* bacteria: its antibacterial effect was similar to that of kanamycin and streptomycin. Complex **I** was also shown to have antibiofilm activity comparable to that of antibiotics ampicillin and ceftriaxone. The cytotoxicity to Vero cells proved to be lower than MIC for *M. luteus*. The ligands of complex **I** were found to have no effects on bacteria or Vero cells growth.

Our present and previous data demonstrate that DNICs represent a promising class of NO donor compounds that can be considered as agents for antimicrobial NO therapy.

## Data Availability

The authors confirm that the data supporting the findings of this study are available within the article and its Appendix A.

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
