# Peer review of "Novel Type of Tetranitrosyl Iron Salt: Synthesis, Structure and Antibacterial Activity of Complex [FeL’2(NO)2][FeL’L”(NO)2] with L’-thiobenzamide and L”-thiosulfate"

_molecules, 2022, doi:10.3390/molecules27206886_

Round 1

Reviewer 1 Report

The article is devoted to a relevant  topic with great practical potential. The article is well-written, in accordance with all the requirements of journal. The only small remark I would note is the Elemental analysis for: C21H21N7O7S5Fe2: Found, % C 33.54, H 2.80, N 12.95, O 14.79, S 21.17, Fe 14.75. 435 Calculated, % C 33.54, H 2.80, N 12.95, O 14.79, S 21.17, Fe 14.75 (Lines 435-436). It also contains an analysis of the oxygen content of the substance. Analysis needs to be checked. Overall, I am impressed with the article and believe that it can be accepted after minor revision.

Author Response

Authors agree with the reviewer’s remark, and the corresponding corrections have been made in the text (line 439).

Reviewer 2 Report

A well-prepared manuscript devoted to the preparation and study of the properties of a new nitrosyl complex, which is an exogenous NO donor, is presented. Considering the prospects of such compounds for application, as well as extensive experimental material, including data from both physicochemical and biological experiments, I believe that the article is useful for a wide range of specialists, and will be interesting to readers and well cited.

There are a number of technical notes on the manuscript:

Scheme 1 is better to give on two lines, increase the size of the arrows and rationally place the captions above them

The title of Figure 8 should be changed to " Various forms of thioamides"

Line 415, in the formula FeSO4•7H2O "4" to make a subscript and then check the entire text for the same remark (line 626, 675, 686, 689).

It should be clarified, when determining Fe AAS, how the sample was prepared, whether mineralization was carried out or a solution of the complex was used.

It would be excellent to supplement the elemental analysis data with HRMS data, if possible.

In the experimental, it is necessary to indicate how the crystal was described for single-crystal X-ray diffraction analysis, what color it has.

For MTT test method, you must provide a references.

Author Response

  1. Scheme 1 is better to give on two lines, increase the size of the arrows and rationally place the captions above them.

Authors agree with the reviewer’s remark, and the corresponding corrections have been made in the text (lines 99-100).

  1. The title of Figure 8 should be changed to "Various forms of thioamides"

Authors agree with the reviewer’s remark, and the corresponding corrections have been made in the text (line 218).

  1. Line 415, in the formula FeSO4×7H2O “ 4” to make a subscript and then check the entire text for the same remark (line 626,675,686,689)

Authors agree with the reviewer’s remark. We checked subscripts in the entire text and the corresponding corrections have been made (lines 416, 473, 628, 677, 688, 690, 692-694, 696, 708, 723).

  1. It should be clarified, when determining Fe AAS, how the sample was prepared, whether mineralization was caried out or solution of complex was used.

Authors agree with the reviewer’s remark, and the corresponding corrections have been made in the text (lines 421, 422).

  1. It would be excellent to supplement the elemental analysis data with HRMS data, if possible.

Unfortunately, at present, authors would not be able to supplement the elemental analysis data with HRMS data.

  1. In the experimental, it is necessary to indicate how the crystal was described for single-crystal X-ray diffraction analysis, what color it has.

The color of crystals was red-brown, we added also the average size of the crystals (line 436).

  1. For MTT test method, you must provide a referenсes.

The authors agree with the reviewer’s remark, and the corresponding corrections have been made in the text (line 558), reference 54 was added (lines 755-756).